# Practical Limits for Two Fundamental Approaches to Designing Particle Size Distributions to Address a Specific Physical Property like Viscosity

**DOI:** 10.3390/polym13183047

**Published:** 2021-09-09

**Authors:** Richard D. Sudduth

**Affiliations:** Materials Research & Processing, LLC, 3718 Dunlin Shore Court, Peachtree Corners, GA 30092, USA; RichSudduth@Earthlink.Net

**Keywords:** packing fraction, viscosity, modulus, impact, suspensions, interaction coefficient, relative viscosity, generalized viscosity model

## Abstract

It has previously been shown that optimum particle size distributions with a maximum packing fraction can be achieved from a straight line plot of the accumulated sum of particle volume fractions versus the square root of particle size. This study addresses practical limits for two dominant fundamental approaches to designing particle size distributions to address the effect on a specific physical property such as viscosity. The two fundamental approaches to obtain such a straight line would include: the first design approach would be generated utilizing the same initial particle size, Dmin, but by using different ultimate particle sizes, Dmax. The second design approach would be generated where each distribution starts with the same initial particle size, Dmin, and ends with the same ultimate particle size, Dmax. The first design approach is particularly useful to identify the possible slopes available based on the smallest and largest particle sizes available. The second design approach can be utilized to identify the preferred ratio between particles, Z, and the number of different particle sizes, n, to be utilized in the final particle blend. The extensive empirical experimental evaluations of particle size distributions generated by McGeary were then utilized to confirm the limits.

## 1. Introduction

Over the years, research involving the blending of different pseudo-spherical particle sizes to maximize the packing fraction has been addressed to optimize several physical properties including viscosity [1,2,3,4,5,6,7,8,9,10,11,12,13,14,15,16,17,18,19,20,21,22,23,24,25], modulus [26,27,28,29,30,31,32,33,34], and impact [35,36,37,38,39,40,41,42]. The importance of the packing fraction for a property such as viscosity can easily be evaluated by addressing one of the more general models to characterize viscosity, such as the following generalized viscosity model [7,8,9,10]:(1)$$\mathrm{Ln}\left(\eta /\eta_{o}\right)=\left(\frac{\left[\eta \right]\varphi_{n}}{\sigma -1}\right)\;\{{\left(\frac{\varphi_{n}-\varphi }{\varphi_{n}}\right)}^{1-\sigma }-1\}\;\mathrm{for}\;\sigma \neq 1$$
(2)$$\sigma =\frac{\lambda_{\mathrm{PC}}}{{\bar{D}}_{1}}+\sigma_{s}$$
where
η = suspension viscosity;η_o_ = viscosity of suspending medium;[η] = intrinsic viscosity;σ = interaction coefficient;λ_PC_ = particle–particle component of the interaction coefficient;σ_s_ = solvent–particle component of the interaction coefficient;φ = particle volume fraction concentration;φ_n_ = particle packing fraction;${\bar{D}}_{1}$ = number average particle size or the first moment average particle size.

While optimizing the packing fraction is important for large-scale pseudo-spherical aggregates such as asphalt applications, there are also many other applications for optimizing particle size distributions on a much smaller scale. Many small-scale particle size analyzers maintain a constant ratio between successive particle sizes analyzed, Z, such as for the LS 130 Coulter Counter where Z = 1.09532. For reference, the maximum range of particles that can be indicated using the LS 130 Coulter Counter is only from 0.1 microns to 899.7 microns. For such a particle size analyzer, it is normally found that each i^th^ particle group of diameter *D*_i_ in a particle distribution can then be evaluated (where, in general, Z > 1) as:(3)$$D_{i}=Z^{i-1}D_{1}$$
where
D_Min_ = D_1_ = diameter of the smallest particle size in the particle distribution.

Several attempts have been made over the years to elucidate the best theoretical [8,9,10,41,43,44,45,46,47,48,49] approach to characterize the blending of different pseudo-spherical particle sizes to maximize the packing fraction. Based on experimental considerations, Kaeuffer [49] found that the optimum particle size distribution for a pigment in a paint would be achieved when the accumulated volume fraction for the distribution was directly proportional to the square root of the particle diameter. In other words, according to Kaeuffer, the accumulated volume fraction from the smallest particle size up to the largest particle size considered, D_β_, was assumed to be dependent on the square root of the particle diameter as:(4)$$\sum_{i=1}^{i=\beta }f_{i}=a\sqrt{D_{\beta }}$$
where:f_i_ = volume fraction of particles of the i^th^ type

Sudduth [8,9,10] was able to extend the experimental observations found by Kaeuffer by showing theoretically that the volume fraction for the i^th^ particle size in an optimum particle size distribution can be calculated as:(5)$$f_{i}=\frac{V_{i}}{V_{T}}=\frac{{N_{i}D}_{i}^{3}}{\sum_{i=1}^{n}{N_{i}D}_{i}^{3}}=\frac{\sqrt{D_{i}^{}}}{\sum_{i=1}^{n}\sqrt{D_{i}^{}}}$$
where:V_i_ = volume of the ith particles;V_T_ = total volume of all particles;f_i_ = volume fraction of particles of the ith type.

Sudduth [44] also found that for volume fractions calculated using Equation (5) with a constant ratio between particle sizes, a straight line of the following form would necessarily result:(6)$$\sum_{i=1}^{i=\beta }f_{i}=a\sqrt{D_{\beta }}+b$$

Note specifically that the straight line described by Equation (6) typically does not necessarily go through the origin.

In addition, as already indicated, several attempts have been made over the years to elucidate the best theoretical [8,9,10,41,43,44,45,46,47,48,49] approach to characterize the blending of different pseudo-spherical particle sizes to maximize the packing fraction. The theoretical analyses then required extensive experimental blending data [50,51,52] to maximize the packing fractions from several different particle size distributions to compare with the theoretical models.

Probably the most extensive empirical study for blending different particle sizes was generated by McGeary [52] to maximize the packing fraction in fuel rods for use in nuclear generated power plants.

The experimental results from McGeary have then be utilized in this study to identify how Equations (5) and (6) can be generated theoretically. The theoretical background used to generate Equations (5) and (6) also resulted in the identification of a ratio of two specific particle size averages that has been particularly successful in indicating the packing effectiveness of any given particle size distribution.

At this point, it is useful to identify how to best develop an optimum distribution described by the straight line requirement, as indicated by Equation (6). This study will address the practical limits for the two dominant fundamental approaches to designing particle size distributions to address a specific physical property such as viscosity. Two fundamental approaches to obtain such a straight line would include:
(1)The first design approach would be generated utilizing the same initial particle size, D_min_, but by using different ultimate particle sizes, D_max_.(2)The second design approach would be generated where each distribution starts with the same initial particle size, D_min_, and ends with the same ultimate particle size, D_max_.

The first design approach is particularly useful to identify the possible slopes available based on the smallest available particle size and the largest available particle size that can be used in the particle blend for a specific application. Once the slope has been identified, the second design approach can be utilized to identify the preferred ratio between particles, Z, and the number of different particle sizes, n, to be utilized in the final particle blend.

## 2. Particle Size Distribution Concepts from an Analysis of McGeary’s Particle Size Distributions

Probably the most extensive particle size blending study for spherical particles was generated by McGeary [52] as a result of his effort to maximize the packing fraction for fuel rods utilized in nuclear generated power plants. The packing fraction for five binary sets of McGeary’s blending data for spherical particles have been plotted in Figure 1 as a function of the volume fraction of the smallest particle. The volume fraction for each particle, f_i_, in Figure 1 can be described as:(7)$$f_{i}=\frac{V_{i}}{V_{T}}=\frac{{N_{i}D}_{i}^{3}}{\sum_{i=1}^{n}{N_{i}D}_{i}^{3}}$$

Taking the ratio of two volume fractions yields:(8)$$\frac{f_{i}}{f_{k}}=\frac{V_{i}}{V_{k}}=\frac{{N_{i}D}_{i}^{3}}{{N_{k}D}_{k}^{3}}$$

Rearranging gives:(9)$$N_{i}=N_{k}\left(\frac{D_{k}^{3}}{D_{i}^{3}}\right)\left(\frac{f_{i}}{f_{k}}\right)$$

Assuming that N_k_ = 1 for largest particle diameter, then the first moment, ${\bar{D}}_{1}$, and the fifth moment, ${\bar{D}}_{5}$, average particle sizes can be calculated as:(10)$${\bar{D}}_{1}=\frac{\sum_{i=1}^{n}N_{i}D_{i}}{\sum_{i=1}^{n}N_{i}}$$
(11)$${\bar{D}}_{5}=\frac{\sum_{i=1}^{n}N_{i}D_{i}^{5}}{\sum_{i=1}^{n}N_{i}D_{i}^{4}}$$
where:V_i_ = volume of the ith particles;V_k_ = volume of the kth particles;V_T_ = total volume of all particles;f_i_ = volume fraction of particles of the ith type;f_k_ = volume fraction of particles of the kth type;${\bar{D}}_{1}$ = number average particle size or the first moment average particle size;${\bar{D}}_{5}$ = fifth moment average particle size;N_i_ = number of particles of the i^th^ particle size;D_i_ = diameter of the i^th^ particle size;n = number of different particle sizes in mixture.

Using Equations (7)–(11), the ratio ${\bar{D}}_{5}$/${\bar{D}}_{1}$ particle size averages for McGeary’s five binary particle size distributions have been plotted in Figure 2 as a function of the volume fraction of the smallest particle. The theoretical maximum value of ${\bar{D}}_{5}$/${\bar{D}}_{1}$ included in each of the binary plots Figure 1 and Figure 2 has previously been shown [8,10] to easily be calculated as:(12)$$f_{\mathrm{imax}}=\frac{\sqrt{D_{i}^{}}}{\sum_{i=1}^{n}\sqrt{D_{i}^{}}}$$

Note that the maximum volume fraction, f_imax_, for component i can be calculated as the square root of diameter i divided by the sum of the square roots of all particle diameters in the distribution.

Of all possible ratios, it has been shown in a previous extensive study [8] that only the ${\bar{D}}_{5}$/${\bar{D}}_{1}$ ratio of average particle sizes can accurately predict both the location and relative magnitude of the maximums for McGeary’s [52] five binary particle size distributions as indicated in both Figure 1 and Figure 2. In a later study [10], it was further established that Equation (12) also applies to the calculation of the volume fraction for any particle size at the maximum packing fraction independent of how many different particles have been included in the particle size distribution.

The maximum packing fraction found by McGeary [52] for several particle size distributions from binary blends to distributions with up to four particles has been summarized in Table 1. Note that, using Equation (12), the calculated volume fractions yielding the maximum packing fractions for McGeary’s distributions in Table 1 were found [8] to be in good agreement with the volume fractions that McGeary found experimentally.

Since monodisperse spherical particles have essentially the same packing fraction, φ_m_, independent of their particle size, then smaller particles can then fit in between the larger particles. In general, Lee [46] has indicated that monodisperse spherical particles have a maximum packing fraction of φ_m_ = 0.589 for loose random packing and φ_m_ = 0.639 for dense random packing. If the assumption is made that smaller spheres will fit into the voids left by the larger spheres, then the ultimate maximum packing fraction, φ_nult_, for n different particle sizes yields the following equation, which has been previously published [8] in detail, has been summarized in Appendix A:(13)$$\varphi_{\mathrm{nult}}=1-{\left(1-\varphi_{m}\right)}^{n}$$

For reference, the ultimate packing fraction has been summarized in Table 2 for up to 12 different particles for both loose random packing and for dense random packing. For reference, it needs to be noted that the maximum possible value for ${\bar{D}}_{5}$ cannot be greater than the magnitude of the largest particle diameter in the distribution and the minimum possible value for ${\bar{D}}_{1}$ cannot be smaller than the magnitude of the smallest particle diameter in the distribution. The ultimate packing fraction, φ_nult_, described by Equation (13) is also shown in Figure 3 along with the maximum packing fractions that were established experimentally by McGeary’s distributions as indicated in Table 1. Note in Figure 3 how amazingly close McGeary’s experimental results were to the ultimate theoretical packing fraction.
(13)$$\varphi_{\mathrm{nult}}=1-{\left(1-\varphi_{m}\right)}^{n}$$

At this point, the maximum packing fraction for any particle size distribution then can be calculated using the following equation, which has been developed in a previous publication [8]:(14)$$\varphi_{n}=\varphi_{\mathrm{nult}}-(\varphi_{\mathrm{nult}}-\varphi_{m})e^{\alpha (1-{({\bar{D}}_{5}/\bar{D}}_{1}))}$$
where:φ_n_ = particle packing fraction;φ_nult_ = ultimate Particle packing fraction;φ_m_ = monodisperse particle packing;α = particle size distribution constant = 0.268;${\bar{D}}_{1}$ = number average particle size or the first moment average particle size;${\bar{D}}_{5}$ = fifth moment average particle size.

This equation for the maximum packing fraction, φ_n_, was required to be a function of the ratio ${\bar{D}}_{5}$/${\bar{D}}_{1}$ to successfully predict the maximums for McGeary’s additional blends for tertiary and quaternary blends as well as indicated in Table 1. It has also been found that Equation (14) has also been successfully extended to include blends with an indefinite number of particles as well.

It has also been found that the accumulated volume fractions at the maximum packing fraction for three of McGeary’s particle size distributions can be plotted as effective straight lines as a function of the square root of the particle diameter as indicated in Figure 4 and Figure 5 as well as in Table 3. The following equations were used to obtain the straight lines indicated in Figure 4 and Figure 5 as the results in Table 3:(15)$$\sum_{i=1}^{i=\beta }f_{i}=a**\;\sqrt{D_{\beta }}+b**$$
(16)$$a**=\frac{1-f_{\min}}{\sqrt{D_{\max}}-\sqrt{D_{\min}}}$$
(17)$$b**=\frac{f_{\min}\sqrt{D_{\max}}-\sqrt{D_{\min}}}{\sqrt{D_{\max}}-\sqrt{D_{\min}}}$$

The correlation between the McGeary’s experimental results and the calculated results in Figure 4 and Figure 5 as well as in Table 3 were quite remarkable. This again suggested that the calculated results appear to be approaching the correct theoretical explanation of McGeary’s experimental results.

Finally, it should be noted that the ratio between McGeary’s particles sizes as indicated in Table 1 ranged from 5.55 to 11.27. Consequently, it was of interest to see if there was an optimum ratio that would be preferred between particle sizes. The resulting derivation of the optimum ratio between particles, which has been generated in a recent publication [43], simply looked at the particle ratio between the maximum packing fraction between n and n+1 particles as described by Equation (13). An outline of this derivation which resulted in the following equation has been summarized in Appendix B:(18)$$\frac{D_{\mathrm{nult}}}{D_{(n+1)\mathrm{ult}}}={\left(\frac{1}{1-\varphi_{m}}\right)}^{2}=Z_{\mathrm{Optimum}}$$

Equation (18), as derived in Appendix B, indicates that the optimum ratio between particles, Z_Optimum_, should be a direct function of the monodisperse packing fraction for spheres, φ_m_. Again, note that Lee [46] found that for loose random monodisperse packing of spheres (φ_m_ = 0.589), whereas for dense random monodisperse packing of spheres (φ_m_ = 0.639). Consequently, the optimum Z results summarized in Table 4 have been calculated for monodisperse packing fractions that range from loose random packing to dense random packing. As indicated in Table 4, values of the optimized ratio between particles would then appear to range between 5.92 and 7.67, which is approximately the same range as that found by McGeary experimentally, as indicated in Table 1. In general, the agreement between theory and McGeary’s experimental results appears to be quite good.
(18)$$\frac{D_{\mathrm{nult}}}{D_{(n+1)\mathrm{ult}}}={\left(\frac{1}{1-\varphi_{m}}\right)}^{2}=Z_{\mathrm{Optimum}}$$

Consequently, it is of interest to include the following quote from the final statement that McGeary [52] made in his conclusion to his study:

“To produce efficient packing, there should be at least a sevenfold difference between sphere diameters of the various individual components. This size difference was shown to be associated with the ‘triangular pore size’ in the established packing through which the next component had to migrate to reach a permanent site.”

This result again indicates that the results that McGeary found experimentally agree very nicely with the theoretical analysis of his results as addressed in this study.

## 3. Influence of the Maximum Packing Fraction on the Generalized Viscosity Model

The generalized viscosity model introduced earlier in Equations (1) and (2) as well as Equations (7)–(14) were developed by Sudduth [7,8,9,10] to describe the viscosity of suspensions and composites with spherical particles as a function of the pigment volume concentration, φ. Three primary variables—the intrinsic viscosity, [η]; the maximum packing fraction, φ_n_; and an interaction coefficient, σ—were included in the original development of the generalized viscosity equation. As indicated in Table 5, the modification of the interaction coefficient, σ, can characterize the solubility of the suspended phase [53,54,55].

The modification of the interaction coefficient, σ, can also be adjusted to yield several different well known suspension equations. For example, when σ = 0, the Arrhenius [23] equation results; when σ = 1, the Krieger–Dougherty [24] equation results; and when σ = 2, the Mooney [25] equation results.

As indicated in Figure 6, an adjustment of the interaction coefficient to yield a change in the suspension solubility would also typically result in a change in the shape of the viscosity curve, as indicated in Figure 6. Note that curves in Figure 6 were all calculated using the Einstein limit [21,22] for the intrinsic viscosity of [η] = 2.5 with a maximum packing fraction of φ_n_ = 0.59.

Similarly, the results in Figure 7 were calculated with a range of values for the maximum packing fraction, again using the Einstein limit for the intrinsic viscosity of [η] = 2.5 but with a constant interaction coefficient of σ = 1.01. The results in Figure 7 then suggest that the viscosity of suspensions can be significantly reduced at higher concentrations by increasing the maximum packing fraction of the particle size distribution. In other words, increasing the maximum packing fraction to reduce viscosity, as indicated in Figure 7, has similar characteristics to decreasing the interaction coefficient to modify the solubility of the suspended particles, as indicated in Figure 6.

## 4. Extension of the Influence of the Slope and Intercept of Straight Line for the Optimum Particle Size Distribution with Improved Blending Considerations

Many particle size analyzers maintain a constant ratio, Z, between successive particle sizes analyzed such as that for the LS 130 Coulter Counter where Z = 1.09532. For such a particle size analyzer, it is normally found that each i^th^ particle group of diameter D_i_ in a particle size distribution can then be evaluated (where, in general, Z >1) as:(19)$$D_{i}=Z^{i-1}D_{1}$$

Similarly,
(20)$$D_{\mathrm{Max}}=Z^{n-1}D_{\mathrm{Min}}=Z^{n-1}D_{1}$$

Rearranging then also gives:
(21)$$Z={\left(\frac{D_{\mathrm{Max}}}{D_{1}}\right)}^{\left(\frac{1}{n-1}\right)}$$
and
(22)$$n=1+\mathrm{Ln}\left(\frac{D_{\mathrm{Max}}}{D_{1}}\right)/\mathrm{Ln}\;(Z)$$
where:
D_Min_ = D_1_ = diameter of the smallest particle size in the particle distribution.

Note that the maximum range of particles that can be indicated using the LS 130 Coulter Counter is from 0.1 microns to 899.7 microns. It can easily be shown using Equation (19) that the maximum number of different particle size groups that can then be analyzed with this machine is n = 101.

As described earlier, Equation (6) can be used to describe an optimum particle size distribution with a straight line as:
(23)$$\sum_{i=1}^{i=\beta }f_{i}=a*\;\sqrt{D_{\beta }}+b*$$

The values for a* and b* in Equation (23) can then be directly calculated based on the following two boundary conditions:

For n = 1, D_β_ = D_1_ = D_Min_ and
(24)$$F_{\Sigma 1}=\sum_{i=1}^{i=1}f_{i}=f_{1}=\frac{\sqrt{D_{1}}}{\sum_{i=1}^{i=n}\sqrt{D_{i}}}$$

For n = β, D_β_ = D_β_ and
(25)$$f_{\Sigma \beta }=\sum_{i=1}^{i=\beta }f_{i}=\frac{\sum_{i=1}^{i=\beta }\sqrt{D_{i}}}{\sum_{i=1}^{i=n}\sqrt{D_{i}}}$$

It has also been shown in Appendix C that the sum $\sum_{i=1}^{n}\sqrt{D_{i}^{}}$ reduces to:(C4)$$\sum_{i=1}^{n}\sqrt{D_{i}^{}}=\sqrt{D_{1}^{}}\left(\frac{\sqrt{Z^{n}}-1}{\sqrt{Z}-1}\right)$$

Combining Equation (C4) as well as Equations (20), (24) and (25) with Equation (23) yields:(26)$$a*=\left(\frac{1}{\sqrt{D_{1}}}\right)\left(\frac{\sqrt{Z}}{\sqrt{Z^{n}}-1}\right)$$
(27)$$b*=\left(\frac{1}{1-\sqrt{Z^{n}}}\right)$$

Note that Equation (26) indicates that the slope for the straight line will always be positive. Conversely, Equation (27) indicates that the value of the straight line intercept, b*, can never be positive. However, recall that for McGeary’s data using Equation (17) the straight line intercept, b**, can be positive if:(28)$$f_{min}\sqrt{D_{\max}}>\sqrt{D_{\min}}$$

At the maximum packing fraction for a particle size distribution, the ratios of two maximum volume fractions can be obtained by combining Equations (8) and (12) to give:(29)$$\frac{f_{i}}{f_{k}}=\frac{V_{i}}{V_{k}}=\frac{{N_{i}D}_{i}^{3}}{{N_{k}D}_{k}^{3}}=\frac{\sqrt{D_{i}^{}}}{\sqrt{D_{k}^{}}}$$

Rearranging gives:(30)$$N_{\mathrm{imax}}=N_{\mathrm{kmax}}\;\frac{D_{k}^{3}}{D_{i}^{3}}\;\left(\frac{f_{i}}{f_{k}}\right)=N_{\mathrm{kmax}}\;\frac{D_{k}^{2.5}}{D_{i}^{2.5}}$$

If N_kmax_ = 1 for the largest particle at the maximum packing fraction, then the relative number of smaller particles for the i^th^ particle size indicated by N_imax_ can be easily calculated. In a previous article [41], it was also shown that at the optimum composition for an optimum distribution that the general particle size average, ${\bar{D}}_{x}$, can calculated as
(31)$${\bar{D}}_{x}=\frac{\sum_{i=1}^{n}N_{i}D_{i}^{x}}{\sum_{i=1}^{n}N_{i}D_{i}^{x-1}}=D_{1}\;\frac{\sum_{i=1}^{n}Z^{\left(x-2.5)(i-1\right)}}{\sum_{i=1}^{n}Z^{\left(x-3.5)(i-1\right)}}$$

Further simplification similar to that described in Appendix C yields:(32)$${\bar{D}}_{x}=\frac{\sum_{i=1}^{n}N_{i}D_{i}^{x}}{\sum_{i=1}^{n}N_{i}D_{i}^{x-1}}=\left(\frac{Z^{\left(x-3.5\right)}-1}{Z^{\left(x-2.5\right)}-1}\right)\left(\frac{Z^{(x-2.5)n}-1}{Z^{(x-3.5)n}-1}\right)D_{1}$$

Consequently, at the optimum composition for an optimum particle size distribution, the averages of ${\bar{D}}_{5}$ and ${\bar{D}}_{1}$ can be easily calculated using Equation (32) as well as Equations (10) and (11) to yield the ratio ${\bar{D}}_{5}$/${\bar{D}}_{1}$. Note that the ratio ${\bar{D}}_{5}$/${\bar{D}}_{1}$ calculated using Equation (32) is dimensionless and is only a function of Z and n.

The limiting ${\bar{D}}_{x}$ particle size average that would be of interest depends primarily on which physical property is desired to be addressed. For example, it has already been indicated that to address the viscosity [7,8,9,10] and/or modulus [26], both the first moment or number average particle size, ${\bar{D}}_{1}$, and the fifth moment average particle size, ${\bar{D}}_{5}$, would be need to be evaluated. However, to address an impact performance [41,42], the surface average particle size, ${\bar{D}}_{3}$, would be of interest and could readily be calculated.

## 5. Influence of the of the Ratio between Particle Sizes, Z, on the Slope, Maximum Packing Fraction, φ_n_, and the Average Particle Size Ratio ${\bar{D}}_{5}$/${\bar{D}}_{1}$

In general, the optimum packing fraction, φ_n_, as described by Equation (14) would be expected to increase with an increase in the magnitude of the ratio of two specific average particle sizes, namely the ratio of ${\bar{D}}_{5}$/${\bar{D}}_{1}$. This study has addressed several new approaches to evaluate the ${\bar{D}}_{5}$/${\bar{D}}_{1}$ ratio. In general, the larger the ratio of ${\bar{D}}_{5}$/${\bar{D}}_{1}$ evaluated from particle size distribution measurements, typically, the better the packing of that particle size distribution. Most instruments that measure particle size distribution characterize the distribution with measured values of the volume fraction, f_i_, and/or the relative numbers, N_i_, for each diameter of the i^th^ particle size, D_i_.

The five examples of optimal straight line particle size distributions illustrated in Figure 8 have been generated using Equations (23), (26) and (27) with each distribution starting at the same initial particle size at D_1_ = D_Min_ = 0.1 microns. The straight line for each distribution in Figure 8 then had just six particles that were generated using a different ratio between particles, Z, and each ended at a different particle size, D_Max_, generated as the sixth particle.

The calculated results for the distributions in Figure 8 are summarized in Table 6 and Figure 9 along with an extra distribution that was not included in Figure 8.

There are several observations of interest in Table 6, including:

Note that as the value of Z was increased, the slope continued to decrease while the ratio of ${\bar{D}}_{5}$/${\bar{D}}_{1}$ continued to increase, as indicated in Figure 9. This result was consistent with the earlier observations with McGeary’s data in Table 1 as well as in Figure 4 and Figure 5.

However, note that while the value of the maximum packing fraction, φ_n_, in Table 6 calculated using equation 14 was quite low at low values of the ratio ${\bar{D}}_{5}$/${\bar{D}}_{1}$ it did increase with an increase in the ratio ${\bar{D}}_{5}$/${\bar{D}}_{1}$ and ultimately approached an upper limit.

Finally, the intercept, b*, also had a tendency to increase with an increase in the value of Z. At this point, the value of the intercept does not appear to be as significant as the change in the value of the slope in trying to optimize the packing fraction since it generally seems to have a value near b* = 0.

To better interpret the results summarized in Table 6 as well as the results in Figure 8 and Figure 9, it is important to review the limits of the ultimate maximum packing fraction, φ_nult_, and the maximum packing fraction, φ_n_, as influenced by the number of different particles, n, in the particle size distribution and the average particle size ratio, ${\bar{D}}_{5}$/${\bar{D}}_{1}$. An initial discussion of the number of different particles, n, on the ultimate maximum packing fraction, φ_nult_, were previously indicated earlier in Table 2 and Figure 3 in association with the experimental data of McGeary. With loose random packing and only 2 particles in the blend then the results in Table 2 and Figure 3 indicated that an upper limit maximum packing fraction of only φ_n_ = 0.8311 can be achieved even with the maximum possible ratio of ${\bar{D}}_{5}$/${\bar{D}}_{1}$.

Figure 10 addresses the relationship between the maximum packing fraction, φ_n_, and the average particle size ratio, ${\bar{D}}_{5}$/${\bar{D}}_{1}$, as a function of the number of particles, n, in the blend. Note in Figure 10 that only lower values of the maximum packing fraction are possible for all particle combinations for up to six particles if the value of ${\bar{D}}_{5}$/${\bar{D}}_{1}$ is less than 10. Reference back to Figure 7 also indicates that if the maximum packing fraction cannot achieve above φ_n_ = 0.6 that a possible improvement in viscosity at higher concentrations is minimized.

Two of the three optimum straight line distributions with six particles included in Figure 11 were previously included in Figure 8. However, the third six-particle optimum distribution in Figure 11 had a very different starting particle size with the remaining five particles being generated with a ratio between particles of Z = 1.0953. The results of the calculations for the three particle size distributions in Figure 11 have been included in Table 7. The two particle size distributions in Figure 11 with the same ratio between particle sizes of Z = 1.0953 and the same number of particles, 6, both ended up with the same value of the ratio of ${\bar{D}}_{5}$/${\bar{D}}_{1}$, even though they had very different starting particle sizes. However, the two distributions in Figure 11 with the same ratio between particles of Z = 1.0953 and the same ratio of ${\bar{D}}_{5}$/${\bar{D}}_{1}$ had significantly different slopes. This is a clear case wherein the distribution with a higher value of ${\bar{D}}_{5}$/${\bar{D}}_{1}$ did not yield a lower slope. These two cases with different slopes and different starting particle sizes indicated that the ratio ${\bar{D}}_{5}$/${\bar{D}}_{1}$ would appear to be considerably more important than the value of Z in controlling the maximum packing fraction.

Nevertheless, the results from Figure 8, Figure 9, Figure 10 and Figure 11 and Table 6 and Table 7 did indicate that it appears to be very desirable to maximize the difference between the minimum size particle, *D*_1_ = *D*_Min_, and the maximum size particle, *D*_Max_, when trying to maximize the ratio of ${\bar{D}}_{5}$/${\bar{D}}_{1}$ and minimize the slope. It would also appear to be desirable to have at least six or more different particles in an optimum blend to be able to have the possibility to maximize the upper limit maximum packing fraction. For such an optimal particle size distribution, an increase in the value of ${\bar{D}}_{5}$/${\bar{D}}_{1}$ normally yields an increase in the maximum packing fraction, φ_n_, as well as a lower slope. However, the range between the minimum particle size and the maximum particle size would be expected to depend significantly on the application of interest as well as other properties that may be desired.

## 6. Effect of a Constant Slope on the Influence of the of the Ratio between Particle Sizes, Z, Maximum Packing Fraction, φ_n_, and the Average Particle Size Ratio ${\bar{D}}_{5}$/${\bar{D}}_{1}$

Once the desired maximum and minimum particle sizes have been generated, the slope of the desired particle size distribution would have been established, as shown by the example in Figure 12. At this point, it is desirable to evaluate the number of different particles to be in the distribution as well as the ratio between particle sizes. The relation between the number of particles and the value of Z to be used is clearly indicated using either Equation (21) or (22) once the maximum and minimum particle sizes have been identified so that all of these combinations have the same slope.

The results indicated in Table 8 mathematically describe the relationship between the number of particles and their corresponding Z values for these straight lines with essentially the same slope as the straight line in Figure 12. The results in Table 8 have identified particle size distributions with the same slope but different combinations between the number of particles, n, and the ratio between particle sizes, Z.

All of the slopes and intercepts in Table 8 were calculated using Equations (21) and (22) after first relating the number of particles, n, and Z using Equations (26) and (27). However, it is interesting that while this same process was used for all the calculations in Table 8, the slopes seem to be decreasing ever so slightly in magnitude as the value of Z is increased. This result has no explanation at this point.

Note that the values of Z in Table 8 have been plotted as a function of the number or particles, n, in Figure 13.

In order to understand the results in Figure 13, it is first important to point out that there is an optimum ratio between particles, as addressed in Appendix B. Appendix B addresses the derivation [43] of the optimum ratio between the ultimate packing fractions described in Appendix A. The final result in Appendix B is Equation (B10), which is:(B10)$$\frac{D_{\mathrm{nult}}^{}}{D_{(n+1)\mathrm{ult}}^{}}={\left(\frac{1}{1-\varphi_{m}}\right)}^{2}=Z_{\mathrm{Optimum}}$$

The calculated results using Equation (B10) as previously discussed and summarized in Table 4 yielded a range of this optimum ratio between particles of 5.9199 ≤ Z_Optimum_ ≤ 7.6734. It is apparent in Figure 13 that the location of these optimum values for Z is where the curve increases sharply.

The maximum packing fractions in Table 8 have also been plotted in Figure 14 as a function of the number of particles in the blends. As indicated in Figure 14, there is a sharp drop in the maximum packing fraction when the number of particles in the distributions drops below 10. It is also important to note in Table 8 that the ratio of ${\bar{D}}_{5}$/${\bar{D}}_{1}$ continues to increase unabated as the number of particles in the distribution decreases from 86 to 2.

Note that optimum Z values of 5.9199 and 7.6734 have been included in both Table 6 and Table 8. In Table 6, these values for Z were both applied to the same number of particles in the blend. However, in Table 8, these same values of Z were applied to a constant slope analysis of the blend. The availability of both of these approaches as outlined in Table 6 and Table 8 can be potentially useful when optimizing a particle size distribution for a specific application.

Another important observation in Table 8 is the change in the volume fraction of small particles as the number of different particles in the blend is decreased. For reference, the volume fraction of particles in Table 8 has also been recalculated as the number of small particles relative to the total number of particles in the blend. Note that the volume fraction of the smallest particles has only increased from a volume fraction of 0.00095 to 0.0204 in going from 86 particles to 2 particles. However, as indicated in both Table 8 and Figure 15, the number fraction of the smallest particles increased from 0.203 to approximately 1.000 as the number of particles decreased from 86 particles to 2 particles.

The increase in the number of small particles can also play a significant detrimental role in modifying a property such as viscosity.

In general, the volume of a sphere vs. can be obtained as:
V_s_ = (4/3) π r^3^(33)
where r = sphere radius.

In addition, the surface area of a sphere, A_s_, can be obtained as:
A_s_ = 4 π r^2^(34)

The ratio of surface to volume is then:
A_s_/V_s_ = 3/r(35)

As the diameter of a sphere becomes smaller, the surface-to-volume ratio of the small particles can contribute dramatically as the number of small particles increase. This can be important since the interaction coefficient, σ, in the generalized viscosity model [7,8,9,10] can be described by Equation (2) as:(2)$$\sigma =\frac{\lambda_{PC}}{{\bar{D}}_{1}}+\sigma_{s}$$
where ${\bar{D}}_{1}$ = number average diameter of particles in the blend.

Consequently, as the number of small particles in the blend increases, the interaction coefficient can increase significantly as a result of an increase in the surface to volume of the particles, which, in turn, can increase the viscosity as indicated in Figure 6. This is then another significant reason to have a large number of different-sized particles in the blend to minimize the possibility of having an increase in viscosity as a result of having too many small particles.

In general, as indicated in Figure 10, a minimum of six particles should be included in the final particle blend to be able to maximize the packing fraction. Utilizing the results in Figure 10 along with the other equations in this study, the possibility of using only six particles to generate an effective optimum particle size distribution has been addressed in Table 9. Note in Table 9 that when only six particles are in the distribution and the ratio between particles is Z = 1.0953, a maximum ratio of ${\bar{D}}_{5}$/${\bar{D}}_{1}$ = 1.099 is obtained. As indicated in Figure 10, a ratio ${\bar{D}}_{5}$/${\bar{D}}_{1}$ = 1.099 would obviously be quite inadequate.

Consequently, based on the results in Figure 10, it would generally be undesirable to generate an optimum particle size distribution if the ratio of ${\bar{D}}_{5}$/${\bar{D}}_{1}$ < 25. Therefore, as indicated in Table 10, a minimum requirement of ${\bar{D}}_{5}$/${\bar{D}}_{1}$ ≥ 25 would be a strongly recommended for an optimum particle size distribution to achieve an effective maximum packing fraction. In general, blends with less than six particles would need to be adjusted extremely carefully to make sure that whatever property (such as viscosity) is addressed, the property will not be adversely affected.

## 7. Blends of Multiple Particle Size Distributions to a Constant Slope to Maximize Packing Fraction, φ_n_

After the maximum and minimum particle sizes have been identified, giving rise to a well-defined slope, the creation of an optimum blend can also be achieved by blending several particle size distributions to that slope, as indicated in Figure 16. Note in Figure 16 that the preferred slope has been designated with a separate straight line to which combinations of other particle size distributions can be matched. A measurement of the goodness of the blend can then be obtained by comparing the least square fit straight line of the blends to the preferred straight line. The Z value in this instance has been determined by the particular particle counter used in the blending process such as the Coulter Counter with Z = 1.0953. Such blends can often be achieved very quickly with a computer program.

The disadvantage of many particle counters is that their value of Z is often limited to a value not much greater than Z = 1. While this approach can be effective when utilized properly, the use of multiple sieves can also be used to improve the value of Z addressed in the particle size analysis.

## 8. Conclusions

This study has identified several practical limits to address the two dominant fundamental approaches to designing optimum particle size distributions with a maximum packing fraction. For a property such as viscosity, if the maximum packing fraction cannot reach above φ_n_ = 0.6, the possibility of lowering the viscosity at higher particulate concentrations is significantly reduced. At maximum packing fractions such that φ_n_ > 0.6, the generalized viscosity model has indicated that the viscosity at higher concentrations can be reduced since a higher packing fraction can act similarly to reducing the interaction coefficient. However, it has also been shown that if there is a large difference in the sizes of the particles, if there are fewer than six particles, and if the ratio of ${\bar{D}}_{5}$/${\bar{D}}_{1}$ is less than 10, then it has been shown that too many small particles can cause the viscosity to increase. An increase in the number of small particles associated with an increase in the surface-to-volume ratio of small particles can play a significant role in modifying or degrading a property such as viscosity.

Based on a detailed analysis of McGeary’s extensive study of several binary, tertiary, and quaternary particle size distributions, it has been found that the maximum packing fraction appears to be fundamentally related to the average particle size ratio ${\bar{D}}_{5}$/${\bar{D}}_{1}$. Two important observations identified from a detailed analysis of McGeary’s extensive study included:
(1)Of all the possible ratios of average particle sizes only the ratio of the fifth moment to the first moment average particle sizes, ${\bar{D}}_{5}$/${\bar{D}}_{1}$, was able to effectively characterize both the relative magnitude and the location of the maximum packing fraction for each of McGeary’s binary, tertiary, and quaternary particle size distributions.(2)The location of the maximum value of the ratio, ${\bar{D}}_{5}$/${\bar{D}}_{1}$, at the maximum packing fraction was obtained when the volume fraction for each particle size in the distribution was calculated from the square root of the particle diameter divided by the sum of the square roots of all the particle diameters in the distribution.

At the maximum packing fraction for an optimum constant ratio particle size distribution, a plot of accumulated particle volume fractions vs. square root of particle diameter will normally yield the following characteristics:

For an optimum particle size distribution with a constant ratio between particle sizes, Z, a plot of the accumulated volume fraction vs. the square root of the particle size will, out of necessity, form a straight line with the following characteristics.
If the value of Z >1, then the slope, a*, will be positive and the intercept, b*, will be negative.The magnitude of the optimum maximum packing fraction is normally increased when the ratio of ${\bar{D}}_{5}$/${\bar{D}}_{1}$ has been increased and the slope of the straight line has been lowered.For the same ratio between particles, Z, and the same number of particles, n, the ratio of ${\bar{D}}_{5}$/${\bar{D}}_{1}$ will normally remain the same independent of the value for the initial or minimum particle size, *D*_min_.For the same ratio between particles, Z, if the particle distribution has the same initial particle size, *D*_min_, and the maximum particle size, *D*_max_, has been increased, the ratio of ${\bar{D}}_{5}$/${\bar{D}}_{1}$ must necessarily increase and the slope will normally decrease.An increase in the ratio between particles, Z, normally results in an increase in the ratio of ${\bar{D}}_{5}$/${\bar{D}}_{1}$ with a decrease in the number of different particle sizes, n, and the relative volume fraction of the smallest particle size will normally necessarily increase.

In general, an optimum particle size distribution can be generated if the volume fractions for a particle size distribution are calculated from the square root for each particle diameter divided by the sum of the square roots of the particle diameters, even if there is not a constant ratio between particles. However, if there is not a constant ratio between particle sizes, then such an optimum particle size distribution may only generate an approximate straight line for the accumulated sum of the volume fractions vs. the square root of the particle size. For such an approximate straight line such as was obtained for McGeary’s experimental data that did not necessarily have a constant ratio between particles, it was found that the intercept b** can be positive when:
$$f_{min}\sqrt{D_{\max}}>\sqrt{D_{\min}}$$

The two dominant fundamental approaches to designing optimum particle size distributions with a maximum packing fraction can be described as:
(1)The first design approach would be generated utilizing the same initial particle size, D_min_, but by using different ultimate particle sizes, D_max_.(2)The second design approach would be generated where each distribution starts with the same initial particle size, Dmin, and ends with the same ultimate particle size, Dmax.

For small-scale particle-size distributions measured with an instrument such as a Coulter Counter, typically, the measured volume fractions are generated with a constant ratio between particle sizes. In this case, once the minimum and maximum particle sizes have been established, the ratio ${\bar{D}}_{5}$/${\bar{D}}_{1}$ can be maximized by blending several particle size distributions to an identified straight line to generate the desired final particle size distribution.

It was also found that there is a theoretical optimum ratio between particles, Z_Optimum_, which appears to range from 5.9199 ≤ Z_Optimum_ ≤ 7.6734. This range of ratios was approximately the same range that McGeary found to be optimum in his study as well. This result indicated that these theoretical calculations agreed nicely with McGeary’s extensive experimental results.

The disadvantage of many particle counters is that their value of Z is often limited to a value not much greater than Z = 1 such as the Coulter Counter with Z = 1.0953. Such a blending approach often requires large numbers of particles to be in the blend to be effective. However, such blends can often be carried out very quickly with a computer program.

While particle counters can be effective when utilized properly, the value of Z can be improved significantly by using multiple sieves to generate optimum particle size distributions.

## Figures and Tables

**Figure 1 polymers-13-03047-f001:**
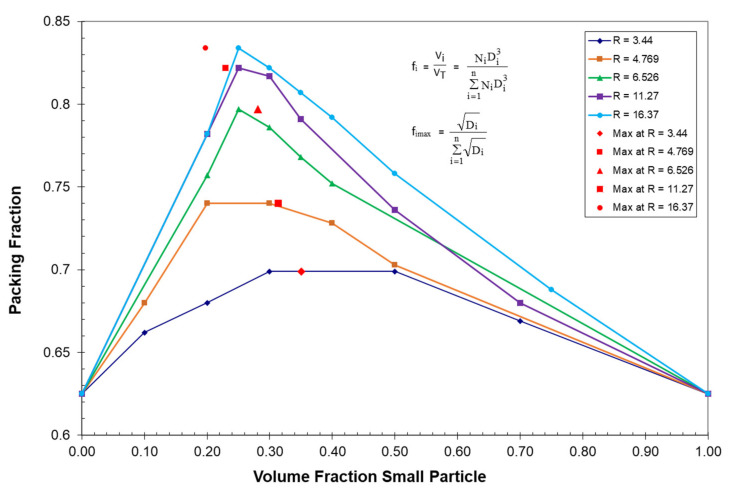
McGeary’s binary particle packing fraction vs. volume fraction small particle.

**Figure 2 polymers-13-03047-f002:**
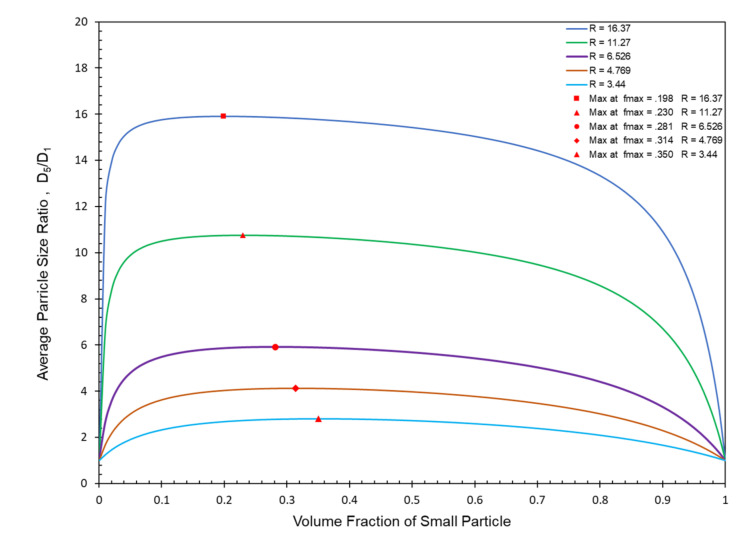
Average particle size ratio, ${\bar{D}}_{5}$/${\bar{D}}_{1}$, for McGeary’s binary particle size ratios vs. volume fraction small particle (where R = D_Large_/D_Small_).

**Figure 3 polymers-13-03047-f003:**
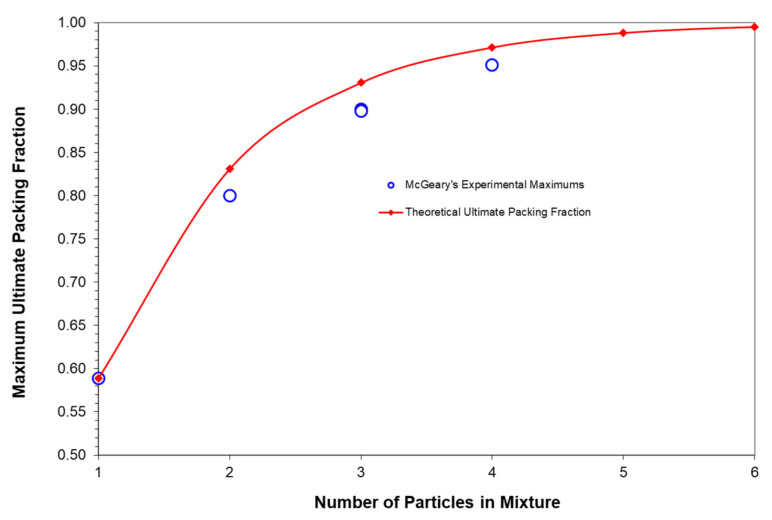
McGeary’s maximums and theoretical maximums vs. number of particles.

**Figure 4 polymers-13-03047-f004:**
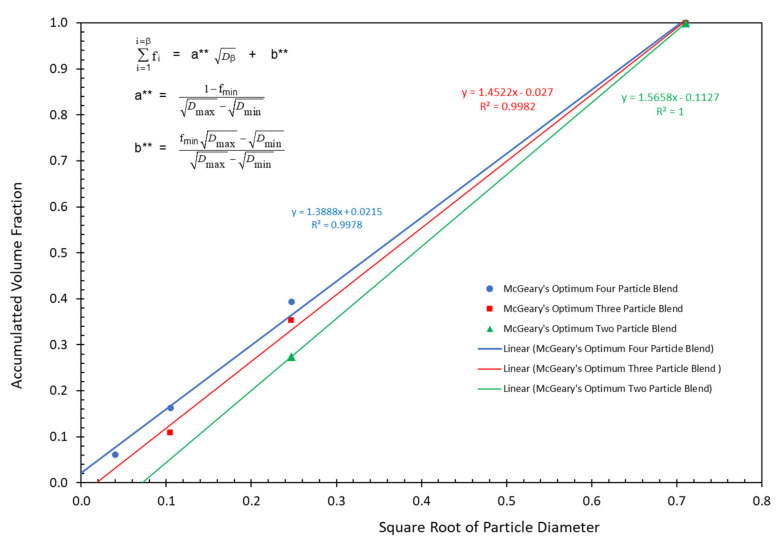
McGeary’s particle blend data vs. square root of particle diameter.

**Figure 5 polymers-13-03047-f005:**
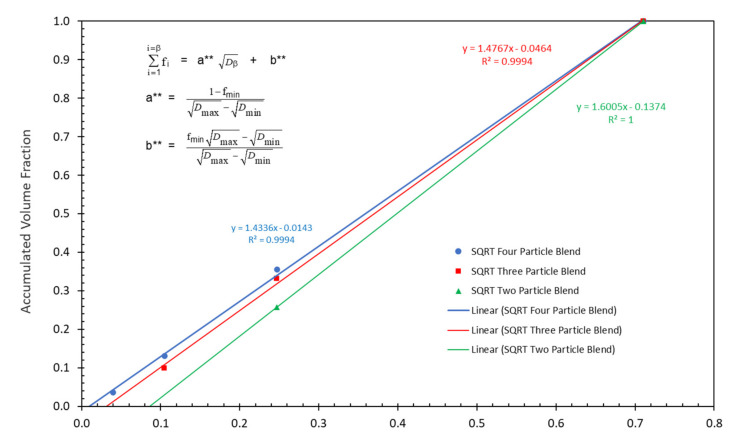
McGeary’s calculated particle blend data vs. square root of particle diameter.

**Figure 6 polymers-13-03047-f006:**
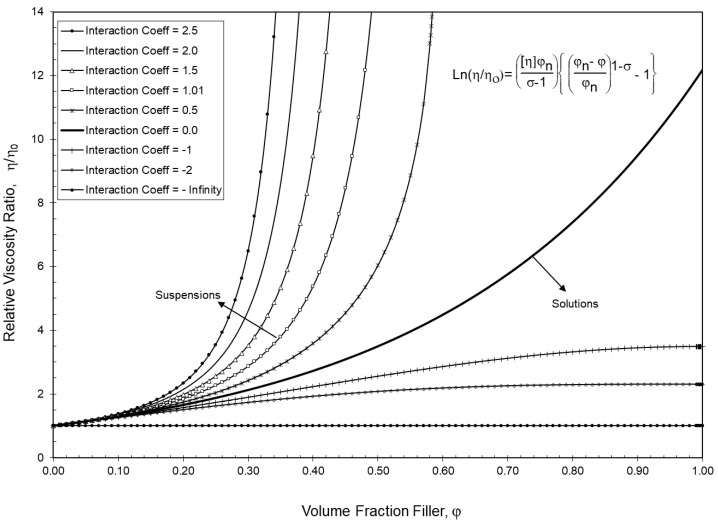
General viscosity model vs. volume fraction filler for a range of interaction coefficients. ([η] = 2.5 and φ_n_ = 0.59).

**Figure 7 polymers-13-03047-f007:**
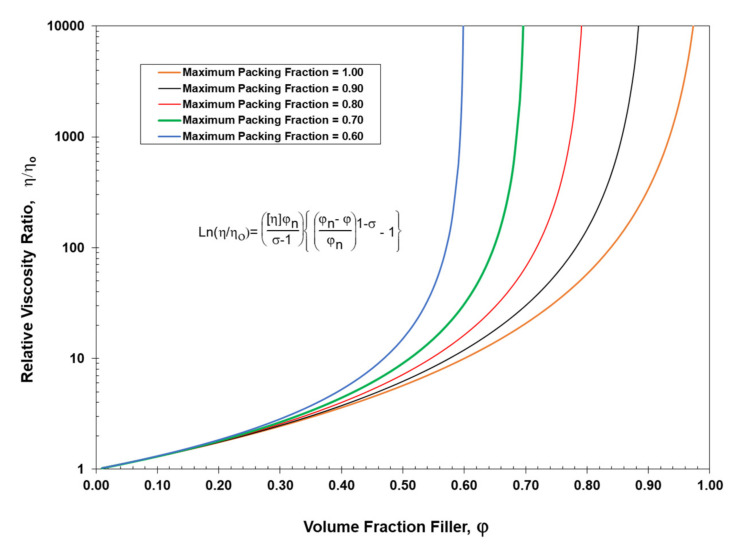
General viscosity model vs. volume fraction filler for a range of maximum packing fractions. ([η] = 2.5 and σ = 1.01).

**Figure 8 polymers-13-03047-f008:**
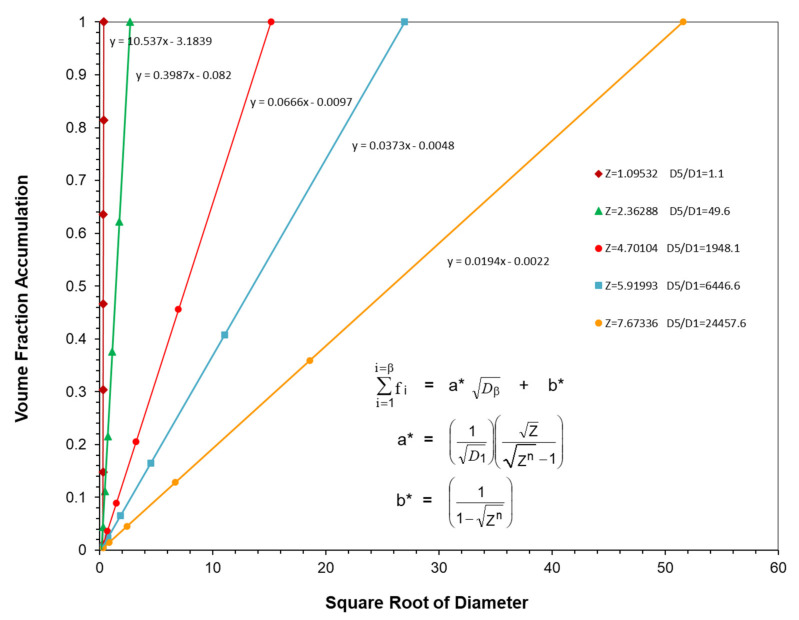
Five calculated optimum straight line distributions with six particles and five different ratios between particles, Z values.

**Figure 9 polymers-13-03047-f009:**
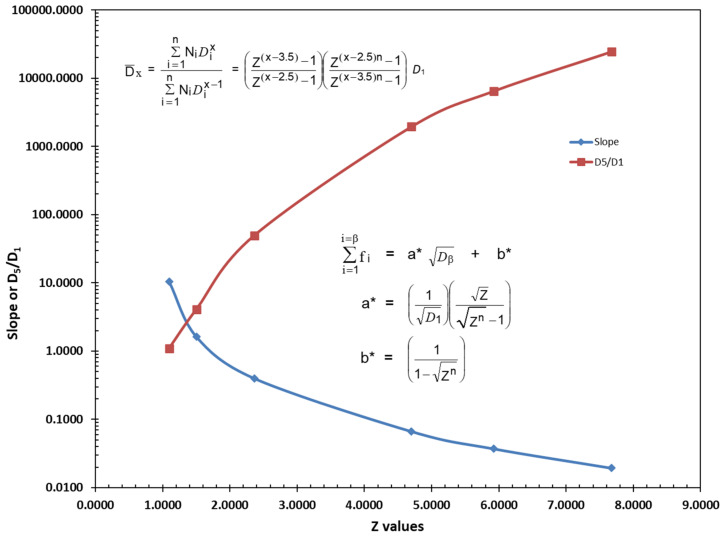
Slope and D_5_/D_1_ vs. 6 different Z values.

**Figure 10 polymers-13-03047-f010:**
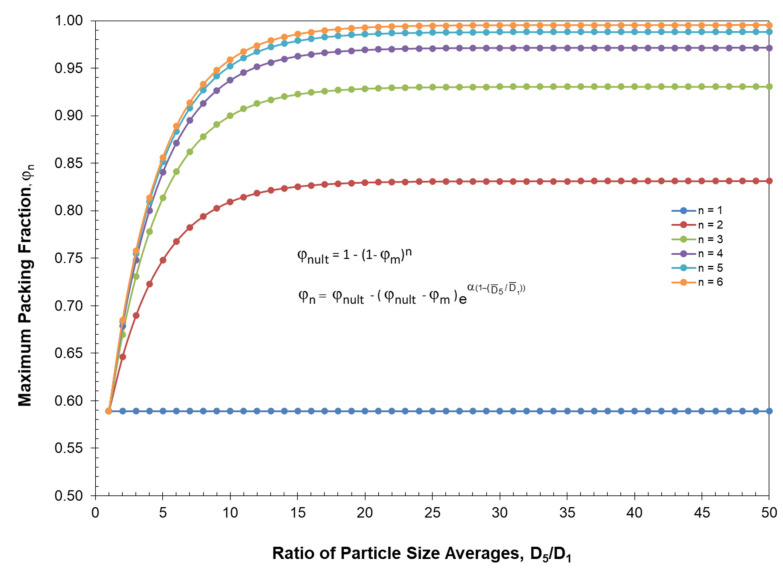
Theoretical maximum packing fraction vs. D5/D1.

**Figure 11 polymers-13-03047-f011:**
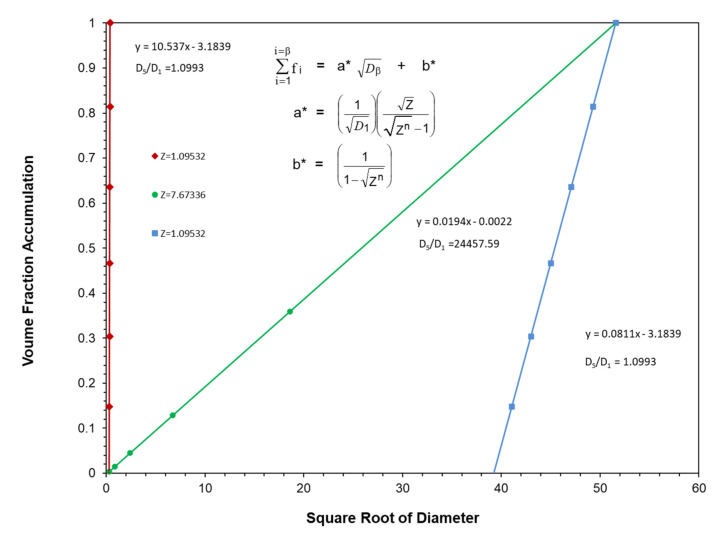
Volume fraction accumulation vs. square root of particle diameter for six particles at two different Z values.

**Figure 12 polymers-13-03047-f012:**
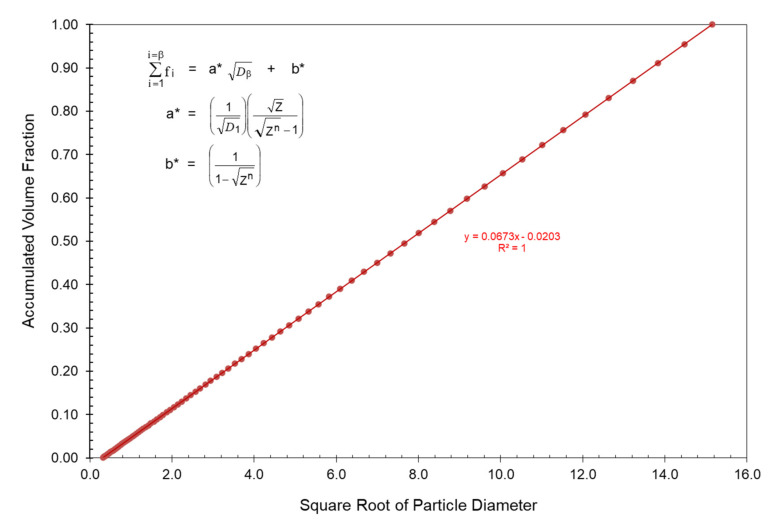
Optimum straight line distribution with the same starting particle and the same ending particle with a constant slope and same ratio between particles (D_min_ = 0.1 and D_max_ = 229.6 and Z = 1.0953).

**Figure 13 polymers-13-03047-f013:**
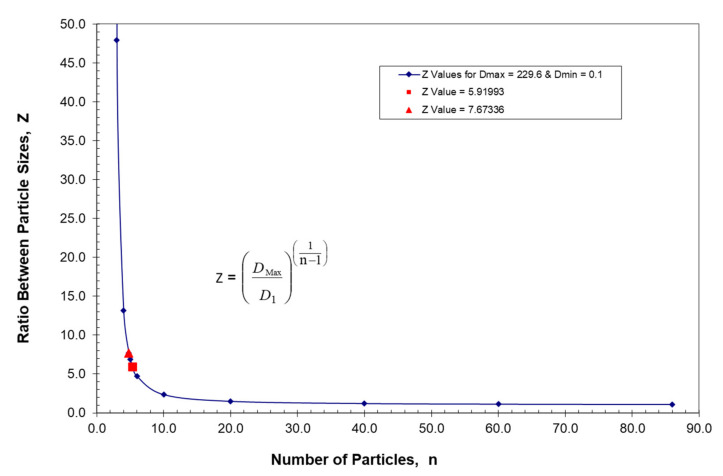
Ratio between particles, Z, vs. number of particles with a constant slope.

**Figure 14 polymers-13-03047-f014:**
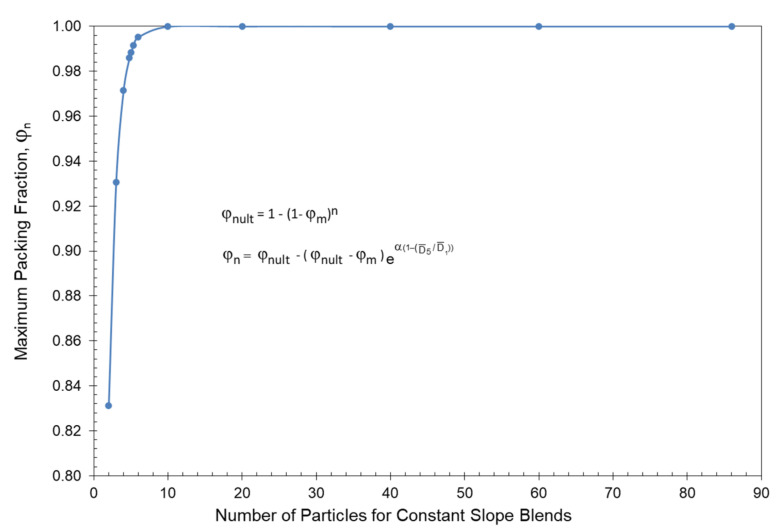
Maximum packing fraction vs. number of different particles in constant slope blends.

**Figure 15 polymers-13-03047-f015:**
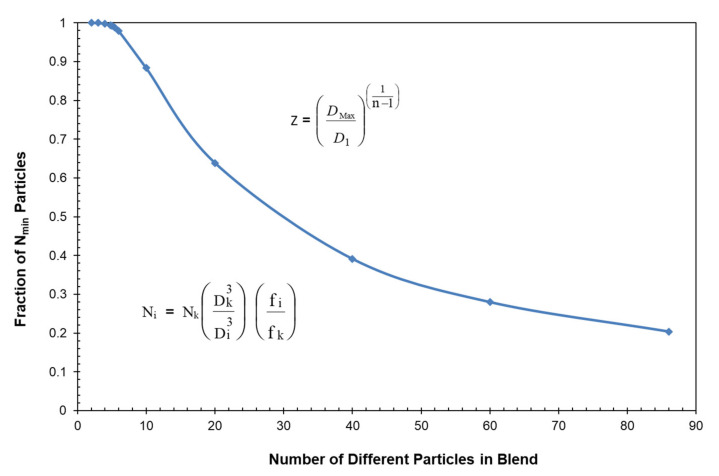
Fraction of number of small particles vs. number of different particles in constant slope blends.

**Figure 16 polymers-13-03047-f016:**
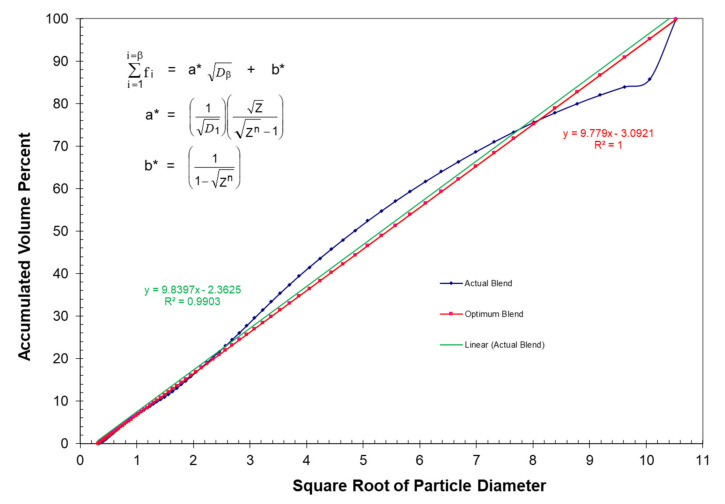
Accumulated volume percent vs. square root of particle diameter for an actual blend of three particle size distributions to the straight line distribution.

**Table 1 polymers-13-03047-t001:** McGeary’s experimental data with calculations from Sudduth [8].

i	Ratio between Particle Diameters	Paticle Diameter	Measured Volume Fraction	Sq Root Calculated Volume Fraction	φ_n_ (Measured)	φ_n_ (Calculated)	φ_ult_ (Theory)	D_5_/D_1_
1	8.2787	0.505	0.607	0.6446				
2	5.5455	0.061	0.230	0.2240				
3	6.875	0.011	0.102	0.0951				
4		0.0016	0.061	0.0363	0.951	0.971	0.971	288.7
1	8.2787	0.505	0.647	0.6688				
2	5.5455	0.061	0.244	0.2325				
3		0.011	0.109	0.0987	0.898	0.931	0.931	41.5
1	11.2727	0.124	0.670	0.7085				
2	6.875	0.011	0.230	0.2110				
3		0.0016	0.100	0.0805	0.900	0.931	0.931	72.1
1	8.2787	0.5050	0.726	0.7421				
2		0.0610	0.274	0.2579	0.800	0.791	0.831	7.7
1		0.5050	1.0000	1.0000	0.580	0.589	0.589	1.0

**Table 2 polymers-13-03047-t002:** Theoretical ultimate packing fraction based on the number of particles for both loose and dense mono-disperse packing fractions.

Number of Particles, n	φ_m_	φ_nult_	Number of Particles, n	φ_m_	φ_nult_
1	0.589	0.5890	1	0.639	0.639
2	0.589	0.8311	2	0.639	0.8697
3	0.589	0.9306	3	0.639	0.953
4	0.589	0.9715	4	0.639	0.983
5	0.589	0.9883	5	0.639	0.9939
6	0.589	0.9952	6	0.639	0.9978
7	0.589	0.998	7	0.639	0.9992
8	0.589	0.9992	8	0.639	0.9997
9	0.589	0.9997	9	0.639	0.9999
10	0.589	0.9999	10	0.639	1.0000
11	0.589	0.9999	11	0.639	1.0000
12	0.589	1.0000	12	0.639	1.0000

**Table 3 polymers-13-03047-t003:** Slope and intercept values for McGeary’s straight line plots compared with the calculated straight line plots.

Measured Volume Fraction		Straight Line Constants Least Square	Straight Line Constants Calculated	Sq Root Calculated Volume Fraction		Straight Line Constants Least Square	Straight Line Constants Calculated
0.607	a**=	1.3888	1.4002	0.6446	a**=	1.4336	1.4370
0.230	b**=	0.0215	0.005	0.224	b**=	−0.0143	−0.0212
0.102				0.0951			
0.061				0.0363			
0.647	a**=	1.4522	1.4709	0.6688	a**=	1.4767	1.4879
0.244	b**=	−0.027	−0.0453	0.2325	b**=	−0.0464	−0.0573
0.109				0.0987			
0.670	a**=	2.835	2.8834	0.7085	a**=	2.923	2.9459
0.230	b**=	0.0069	−0.0153	0.211	b**=	−0.027	−0.0374
0.100				0.0805			
0.726	a**=	1.5668	1.5658	0.7421	a**=	1.6005	1.6005
0.274	b**=	−0.1127	−0.1127	0.2579	b**=	−0.1374	−0.1374

**Table 4 polymers-13-03047-t004:** Optimum ratio between particle sizes.

φ_m_	1/(1-φ_m_)	D_n_/D_n+1_	
0.589	2.433	5.9199	Maximum Monodisperse Loose Random Packing
0.590	2.439	5.9488	
0.595	2.469	6.0966	
0.600	2.500	6.2500	
0.605	2.532	6.4092	
0.610	2.564	6.5746	
0.615	2.597	6.7465	
0.620	2.632	6.9252	
0.625	2.667	7.1111	
0.630	2.703	7.3046	
0.635	2.740	7.5061	
0.637	2.755	7.5890	
0.639	2.770	7.6734	Maximum Monodisperse Dense Random Packing

**Table 5 polymers-13-03047-t005:** Generalized suspension viscosity equation for selected values of the interaction coefficient, σ.

Interaction Coefficient σ	State of Mixture	Simplified Form of Generalized Equation for Equation Derivation	Previous References
−2	Solution	$\;\mathrm{Ln}\left(\eta /\eta_{o}\right)=\left(\frac{\left[\eta \right]\varphi_{n}}{3}\right)\;\left\{{\left(\frac{\varphi }{\varphi_{n}}\right)}^{3}-3{\left(\frac{\varphi }{\varphi_{n}}\right)}^{2}+3\left(\frac{\varphi }{\varphi_{n}}\right)\right\}$	
−1	Solution	$\mathrm{Ln}\left(\eta /\eta_{o}\right)=\left(\frac{\left[\eta \right]\varphi_{n}}{2}\right)\;\left\{2\left(\frac{\varphi }{\varphi_{n}}\right)-{\left(\frac{\varphi }{\varphi_{n}}\right)}^{2}\right\}$	
0	Solution	$\mathrm{Ln}\left(\eta /\eta_{o}\right)=\left(\left[\eta \right]\varphi_{n}\right)\left(\frac{\varphi }{\varphi_{n}}\right)=\left[\eta \right]\varphi \;$	Arrhenius (1887–1917)
	Plastisizer/Intermediate		
1	Suspension	$\mathrm{Ln}\left(\eta /\eta_{o}\right)=\left(-\left[\eta \right]\varphi_{n}\right)\mathrm{Ln}\;\left\{1-\left(\frac{\varphi }{\varphi_{n}}\right)\right\}$	Krieger-Dougherty (1959)
2	Suspension	$\mathrm{Ln}\left(\eta /\eta_{o}\right)=\left(\left[\eta \right]\varphi_{n}\right)\left(\frac{\varphi }{\varphi_{n}}\right)\;{\left\{1-\left(\frac{\varphi }{\varphi_{n}}\right)\right\}}^{-1}=\left(\frac{\left[\eta \right]{\varphi \varphi }_{n}}{\varphi_{n}-\varphi }\right)$	Mooney (1951)
3	Suspension	$\mathrm{Ln}\left(\eta /\eta_{o}\right)=\left(\frac{\left[\eta \right]\varphi_{n}}{2}\right)\;\left\{2\left(\frac{\varphi }{\varphi_{n}}\right)-{\left(\frac{\varphi }{\varphi_{n}}\right)}^{2}\right\}{\left\{1-\left(\frac{\varphi }{\varphi_{n}}\right)\right\}}^{-2}$	

**Table 6 polymers-13-03047-t006:** Summary of straight lines for six optimum particle distributions with different Z values but the same starting particle size.

					φ_m_	0.5890	0.5890
					n =	6	6
					α =	0.2680	0.2680
**Z**	**Slope**	**Intercept**	**D_min_**	**D_max_**	**D_5_/D_1_**	**φ_mult_**	**φ_n_**
1.0953	10.5370	−3.1839	0.1000	0.1577	1.0993	0.9952	0.5997
1.5028	1.6195	−0.4178	0.1000	0.7664	4.1162	0.9952	0.8190
2.3629	0.3987	−0.0820	0.1000	7.3656	49.6001	0.9952	0.9952
4.7010	0.0666	−0.0097	0.1000	229.5989	1948.0385	0.9952	0.9952
5.9199	0.0373	−0.0048	0.1000	727.0811	6446.6039	0.9952	0.9952
7.6734	0.0194	−0.0022	0.1000	2660.2832	24,457.5882	0.9952	0.9952

**Table 7 polymers-13-03047-t007:** Summary of straight lines for three optimum particle distributions with different Z values and different starting particle sizes.

					φ_m_ =	0.5890	0.5890
					n =	6	6
					α =	0.2680	0.2680
**Z**	**Slope**	**Intercept**	**D_min_**	**D_max_**	**D_5_/D_1_**	**φ_mult_**	**φ_n_**
1.0953	10.5370	−3.1839	0.1000	0.1577	1.0993	0.9952	0.5997
7.6734	0.0194	−0.0022	0.1000	2660.3	24,457.6	0.9952	0.9952
1.0953	0.0811	−3.1839	1687.4184	2660.3	1.0993	0.9952	0.5997

**Table 8 polymers-13-03047-t008:** Constant slope maximum packing fractions as a function of number of particles and the ratio between particles, Z.

D_min_ =	0.100					φ_m_ =	0.589	0.589
D_max_ =	229.60					α =	0.2680	0.2680
**Number of Particles n**	**Ratio Between Particles Z**	**Calculated Slope**	**Volume Fraction D_max_**	**Volume Fraction D_min_**	**Fraction of N_min_ Particles**	**D_5_/D_1_**	**φ_mult_**	**φ_n_**
86	1.0953	0.06734	0.0454	0.0009	0.2036	902.86	1.0000	1.0000
60	1.1402	0.06731	0.0647	0.0014	0.2796	937.04	1.0000	1.0000
40	1.2195	0.06727	0.0963	0.002	0.3911	994.89	1.0000	1.0000
20	1.5028	0.06714	0.1874	0.0039	0.6388	1176.07	1.0000	1.0000
10	2.3629	0.0669	0.3543	0.0074	0.8835	1544.80	0.9999	0.9999
6	4.7010	0.06664	0.5440	0.0114	0.9791	1948.05	0.9952	0.9952
5.3518	5.9199	0.06657	0.5941	0.0124	0.9883	2035.73	0.9914	0.9914
5	6.9222	0.06652	0.6249	0.0130	0.9921	2083.71	0.9883	0.9883
4.7978	7.6734	0.06650	0.6439	0.0134	0.9939	2110.85	0.9860	0.9860
4	13.1924	0.06638	0.7289	0.0152	0.9984	2208.15	0.9715	0.9715
3	47.9166	0.06620	0.8581	0.0179	0.9999	2282.46	0.9306	0.9306
2	2296.0000	0.06602	0.9796	0.0204	1.0000	2295.96	0.8311	0.8311

**Table 9 polymers-13-03047-t009:** Estimated limits of the ratio of the maximum diameter/minimum diameter particles in a particle size distribution.

Number of Particles n	Ratio Between Particles Z	D_max_/D_min_	D_5_/D_1_
6	1.09532	1.577	1.099
6	2.00000	32.000	19.809
6	3.00000	243.000	181.0
6	4.00000	1024.000	835.4
6	5.00000	3125.000	2686.2
6	5.91993	7270.793	6446.6
6	6.00000	7776.000	6909.6
6	7.00000	16,807.000	15,275.8
6	7.67336	26,602.849	24,457.6

**Table 10 polymers-13-03047-t010:** Practical limits of the ratio of the maximum diameter/minimum diameter particles in a particle size distribution.

Number of Particles n	Ratio Between Particles Z	D_max_/D_min_	D_5_/D_1_
47	1.09532	65.900	25.997
7	2.00000	64.000	39.519
6	3.00000	243.000	181.0
6	4.00000	1024.000	835.4
6	5.00000	3125.000	2686.2
6	5.91993	7270.793	6446.6
6	6.00000	7776.000	6909.6
6	7.00000	16,807.000	15,275.8
6	7.67336	26,602.839	24,457.6

## Data Availability

The data presented in this study are available on request from the corresponding author.

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
