# Peer review of "Practical Limits for Two Fundamental Approaches to Designing Particle Size Distributions to Address a Specific Physical Property like Viscosity"

_polymers, 2021, doi:10.3390/polym13183047_

Round 1

Reviewer 1 Report

The article is a theoretical consideration and undoubtedly the description contained therein contributes to the development of science. However, some corrections need to be made before publication.
The literature does not include items from recent years, there is only one item from 2019. This should be completed.

The method of citing literature is often grouped, e.g., line 32 (1-25); (26-34); (35-42). The introduction should be expanded with the information contained in these literature items, but also with contemporary publications.

In the article, the application aspect of the given considerations is missed. Please give some additional information.
Not all coefficients in the equations have been described under these equations, for example in the formula 1 the D1 notation, previously not described in the text.

It is necessary to work on the form of writing the mathematical equations because in the current version of pdf not all of them are correctly written.

The R factor (fig. 2) should also be defined in the text.

In table 1 there is information about the measurements. This information should be better described in the text.

Is the description of the x axis in fig 3 correct?

Author Response

  • Response to Reviewer 1
  • 1. Most of the equations in this article needed to be corrected since they were in an unreadable form.  Essentially the corrections involved changing the spacing between the different sections of the equations to get them in the same column as the text.
  • 2. Other corrections were made in the text and included in Red.
  • 3. Some of the Figures, Tables and Text were rearranged to make the Text more readable without interference from the Figures and Tables.
  • 4. The first five articles referred to in the introduction ranged from 2016 to 2018 and two of those articles were review articles from 2017 and 2018. In total this article included 9 articles that were from 2011 to 2019 and an additional seven articles from 2000 -2010. In total the introduction included references to 42 articles.  Consequently, the author feels that these references were adequate and there appears to be no need to add any more references since this article is not a review article.
  • 5. The second paragraph in the introduction lists the application of this article to large scale pseudo spherical aggregates like asphalt applications. However, other smaller particle size applications that could use the results in this article for applications like pigments in paint were mentioned in the third paragraph in the introduction
  • 6. The designation for has now been included for Equation 2.
  • 7. The designation of R as the ratio of the large to small particle has now been included in the caption for Figure 2.
  • 8. The results in Table 1 were adequately defined in Equations 10-14.
  • 9.The description of the X axis in Figure 3 is correct.

Reviewer 2 Report

Two approaches have been discussed in the manuscript that addresses practical limits for the 11 two dominant fundamental approaches to designing particle size distributions thus the viscosity. Only one question to the authors is that if there is other models can do the same prediction here? How is your model compared with them? 

Author Response

  • Response to Reviewer 2

  • 1. Most of the equations in this article needed to be corrected since they were in an unreadable form. Essentially the corrections involved changing the spacing between the different sections of the equations to get them in the same column as the text.
  • 2. Other corrections were made in the text and included in Red.
  • 3. Some of the Figures, Tables and Text were rearranged to make the Text more readable without interference from the Figures and Tables.
  • 4. Some of the articles that have competing models have been mentioned in the text. However, none of these other models has been able to give as effective or as applicable results as the model presented in this article.

Reviewer 3 Report

The manuscript needs to be restructured. In its current form, it is too long. It includes 16 figures and 10 tables, which also need to be better edited. The equations are out of bounds and moved. Abbreviations should be described in the abstract. In general, the manuscript does not fully adhere to the journal's author guidelines.

Author Response

  • Response to Reviewer 3

  • Most of the equations in this article needed to be corrected since they were in an unreadable form. Essentially the corrections involved changing the spacing between the different sections of the equations to get them in the same column as the text.
  • Other corrections were made in the text and included in Red.
  • Some of the Figures, Tables and Text were rearranged to make the Text more readable without interference from the Figures and Tables.
  • The abbreviations in the abstract have been defined adequately with the minimum space available.
  • The author has no plans to decrease the length of this article and in fact has gone to great lengths to minimize the complications presented in this article.

Round 2

Reviewer 1 Report

The authors improved the article, but they have not taken into account all suggestions. 

in the Introduction section, there are still references in grouped manner, e.g., line 32 (1-25); (26-34); (35-42). The introduction should be expanded with the information contained in these literature items. There is no need to place such many references in one place, it is enough to put 2-3 of them in one place not 25 of them as it is in the case eg viscosity. But there is a need to place the contemporary publications. 

Author Response

Author’s responses to the above Reviewer comments:

  • While many viscosity references [1-25] were included in the Introductions section, two of the references were review articles published in 2017 and 2018 and this article was not intended to be a review article.
  • In addition, while some of these 25 references were further elucidated in the Introduction a large number of these articles were indeed elucidated further in the section 3 that specifically addressed the property viscosity. While these references were specifically introduced in the introduction it was not felt appropriate to elucidate them further in the introduction since the separate section addressed the situation more appropriately.

Reviewer 3 Report

The manuscript has been improved, but the author has not considered all suggestions. The manuscript still does not fully adhere to the journal's author guidelines. The abstract must be a single paragraph of about 200 words maximum. References should be indicated in square brackets. Tables must be edited; they may have a footer as indicated in Instructions for Authors. The introduction section must be improved. There are many references grouped. 
The conclusions section is too long. This section must contain the main conclusions or interpretations.

Author Response

Author’s responses to the above Reviewer comments:

  • The abstract was reduced to 201 words.
  • All the references in the text were modified to be in square brackets.
  • The tables were found to be all in order and no change was made to the tables.
  • Some of the Figures, Tables and Text were rearranged to make the Text more readable without interference from the Figures and Tables.
  • While many viscosity references [1-25] were included in the Introductions section, two of the references were review articles published in 2017 and 2018 and this article was not intended to be a review article.
  • In addition, while some of these 25 viscosity references were further elucidated in the Introduction a large number of these articles in the introduction were indeed elucidated further in the section 3 that specifically addressed the property viscosity. While these references were specifically introduced in the introduction it was not felt appropriate to elucidate them further in the introduction since the separate section addressed the situation more appropriately.
  • Most articles include a broad range of references that are often grouped in the introduction. In general, this is often a common practice and it is not clear how this is a problem.   In general, the later sections of many papers then reintroduce some of the articles included in the introduction to elucidate specific points.   In addition, later sections of articles may introduce other references that may relate back to one or more subjects that were mentioned in the introduction.
  • As indicated in the introduction, the contents of this article were intended to be used as a guide to pick particle size distributions appropriately for a broad range of applications that may have significantly different requirements. For example coatings or paints are often involved with particles in the micron range and use instruments where the ratio between particles is just over 1.0 as for example the Coulter counter that has a Z value of approximately 1.095.   On the other hand, optimizing the aggregates for asphalt often use a ratio between particles of Z=2 or more with a maximum particle size that can go up to ¾ inch in size.   Consequently, aggregate gradations of particles for asphalt applications typically use sieves to isolate the desired particle size distribution.
  • Consequently, the conclusion for this article was intended to cover the full range of applications for optimizing particle size distributions. In order, to cover this range the conclusion was reduced as much as possible the address this situation.  Therefore, this conclusion will not be reduced further since it has already been minimized to address the range of particle size distributions that can be encountered.